# Using the Method of “Optical Biopsy” of Prostatic Tissue to Diagnose Prostate Cancer

**DOI:** 10.3390/molecules26071961

**Published:** 2021-03-31

**Authors:** Dmitry N. Artemyev, Vladimir I. Kukushkin, Sofia T. Avraamova, Nikolay S. Aleksandrov, Yuri A. Kirillov

**Affiliations:** 1Laser and Biotechnical Systems Department, Samara National Research University, 443086 Samara, Russia; artemyevdn@ssau.ru; 2Laboratory of Non-Equilibrium Electronic Processes, Institute of Solid State Physics Russian Academy of Sciences, 142432 Chernogolovka, Russia; 3Department of Pathological Anatomy, The First Sechenov Moscow State Medical University under Ministry of Health of the Russian Federation, 119146 Moscow, Russia; studenechek@mail.ru (S.T.A.); dr.klauss@mail.ru (N.S.A.); 4Laboratory of Clinical Morphology, Research Institute of Human Morphology, 117418 Moscow, Russia; youri_kirillov@mail.ru

**Keywords:** prostate cancer, optical biopsy, Raman spectroscopy, diagnosis, partial least squares discriminant analysis

## Abstract

**Simple Summary:**

Analytical discrimination models of Raman spectra of prostate cancer tissue were constructed by using the projections onto latent structures data analysis (PLS-DA) method for different wavelengths of exciting radiation—532 and 785 nm. These models allowed us to divide the Raman spectra of prostate cancer and the spectra of hyperplasia sites for validation datasets with the accuracy of 70–80%, depending on the specificity value. Meanwhile, for the calibration datasets, the accuracy values reached 100% for the excitation of a laser with a wavelength of 785 nm. Due to the registration of Raman “fingerprints”, the main features of cellular metabolism occurring in the tissue of a malignant prostate tumor were confirmed, namely the absence of aerobic glycolysis, over-expression of markers, and a strong increase in the concentration of cholesterol and its esters, as well as fatty acids and glutamic acid.

**Abstract:**

The possibilities of using optical spectroscopy methods in the differential diagnosis of prostate cancer were investigated. Analytical discrimination models of Raman spectra of prostate tissue were constructed by using the projections onto latent structures data analysis(PLS-DA) method for different wavelengths of exciting radiation—532 and 785 nm. These models allowed us to divide the Raman spectra of prostate cancer and the spectra of hyperplasia sites for validation datasets with the accuracy of 70–80%, depending on the specificity value. Meanwhile, for the calibration datasets, the accuracy values reached 100% for the excitation of a laser with a wavelength of 785 nm. Due to the registration of Raman “fingerprints”, the main features of cellular metabolism occurring in the tissue of a malignant prostate tumor were confirmed, namely the absence of aerobic glycolysis, over-expression of markers (FASN, SREBP1, stearoyl-CoA desaturase, etc.), and a strong increase in the concentration of cholesterol and its esters, as well as fatty acids and glutamic acid. The presence of an ensemble of Raman peaks with increased intensity, inherent in fatty acid, beta-glucose, glutamic acid, and cholesterol, is a fundamental factor for the identification of prostate cancer.

## 1. Introduction

Modern methods of diagnosing and treating malignant tumors allow for some optimization of the techniques of their early detection. However, these technologies have had an insignificant effect on the structure of mortality of the working-age population caused by oncologic diseases [1] and, in particular, the prostate cancer (PC) and related complications. In 2018, there were reported 1.276 million new cases of prostate cancer worldwide [2]. It has also been observed that, after the age of forty, the risk of the disease continues to increase with each following year [3]. This type of malignancy is traditionally considered to be among the most challenging to diagnose. As it often develops alongside the hyperplastic and preneoplastic processes, or in connection with them, the visualization of tumor nodules during an ultrasound or radiology examination is obscured. Although in most cases, a conventional morphologic methodology is quite capable of solving the problem of PC confirmation, its routine procedures need some considerable improvement [4]. Therefore, the development of refined, innovative methods of PC diagnosis employing sensitive high-tech instrumentation has become one of the top-priority fields in medical science.

Today, advanced techniques of performing an “optical biopsy” of the tissue, with the aid of Raman and luminescence spectroscopy, are beginning to be used as noninvasive means of diagnosing various diseases and pathological conditions, including tumors of different localizations [5,6,7,8]. In essence, Raman spectroscopy relies on the laser radiation interacting with the tissue, where the illuminated sample scatters the light, thereby altering its initial frequency. Consequently, the frequency shift measured by the spectrometer designates vibrational and rotational excitations of structural elements of the tissue under investigation [9]. Thus, each substance constituting the tissue is characterized by a set of Raman lines with certain spectral positions and fixed relative intensities. It is this unique combination of spectral parameters—Raman molecular “fingerprint”—that enables the detection of variations in cellular metabolism, based on Raman light-scattering intensity. Since tumor development is accompanied by structural and biochemical modifications of the tissue, by registering these changes, Raman spectroscopy makes possible identification of various morphological forms of PC, as well as concurrent hyperplastic or preneoplastic processes. Numerous studies have proven this method of examining the prostatic tissue [10,11,12,13,14] and blood plasma [15] potentially viable in diagnosing prostate cancer.

For example, the work of Dinesh K. R. Medipally [15] demonstrated the practical feasibility of diagnosing prostate cancer by means of Raman spectroscopy of blood plasma at 532 nm excitation wavelength, together with mathematical post-processing of multivariate data. In a review by Rachel E. Kast [10], devoted to Raman spectroscopy application in prostate cancer diagnosis, there are described particular tissue metabolites that produce a resonant optical response to 532 nm laser excitation. These include cytochrome (resonance Raman), localized in cell mitochondria, and the basic fluorochromes of the tissue—the sources of fluorescence, such as NAD(P)H, collagen, retinol, riboflavin, folic acid, pyridoxine, tyrosine, glycation products, tryptophan, melanin, and hemoglobin.

In other scientific works [11,12,13,14], the possibilities of Raman spectroscopy with excitation at different wavelengths of laser radiation (633, 785, and 1064 nm) were investigated. In this works, the differences in the average spectra of cancer and control groups were analyzed, or the method of principal components was applied. In our work, for data analysis, we used the method of projections onto latent structures (PLS) (principal component analysis (PCA) was used at the preliminary stage), which does not show the greatest differences of data between classes, but the most informative bands for separating classes (i.e., it is not necessary that there are maximum differences in the intensities of the spectra). Based on this analysis, the most informative Raman bands for separation (VIP-Variables Importance in Projection) were identified, which were subsequently interpreted as possible biochemical (molecular) and structural changes in tissues.

The objective of the presented investigation is to explore the possibilities of applying optical spectroscopy methods in the differential diagnosis of prostate cancer.

## 2. Results and Discussion

### 2.1. Histological Study

Macroscopic examination of the prostate cancer samples indicated that, in all instances, it appeared as a dense whitish fibrous node without sharply defined boundaries, located on the periphery of the gland, 1.0–10.5 cm in diameter, and an average size of 2.5 cm. As a result of the preliminary standard histological study, acinar adenocarcinoma was confirmed in every PC case. The immunohistochemical study in the regions of acinar adenocarcinoma indicated positive cytoplasmic expression of Alpha-Methylacyl-CoA Racemase (AMACR) markers in epithelial structures, from moderate, at 12.3%, to high, at 87.7% (Figure 1a). The expression of high-molecular-weight cytokeratin (HMWCK), on the other hand, was negative in all PC specimens (Figure 1b). The examined group contained the following stages (TNM): pT2a-27 (41.5%), pT2b-7 (10.7%), pT2c-9 (13.8%), pT3a-15 (23.1%), and pT3b-7 (10.7%). The summary results of clinical and morphological studies are presented in Table 1.

On a macroscopic scale, the tissue removed through transurethral resection of the prostate gland contained numerous nodular areas of pinkish-gray color, with thick consistency anda fibrous structure in cross-section, and that are 3–14 cm in size. On standard histological investigation, the hyperplasia nodes were observed to have substantially increased acinar formations, tightly joined to each other, surrounded with wide regions of coarse fibrous connective tissue. In 92.5% of the cases, it corresponded to benign hyperplasia. The immunohistochemical examination of hyperplasia areas indicated high expression of HMWCK by basal cells and negative expression of AMACR, as shown in Figure 1c,d.

In some cases, glandular structures with the absence of ordered cell layers of the epithelium, signs of nuclear atypia, and a preserved basal membrane were determined in the foci of the study, which was regarded as a high-grade PIN (Prostatic Intraepithelial Neoplasia) (7.69%). However, due to the small number of such sites, the result of their spectral analysis was not of interest for the present study due to the lack of convincing and reliable data (*p* > 0.05).

### 2.2. Raman-Luminescence Spectroscopy with Laser Excitation at 785 nm Wavelength

Given the laser excitation at 785 nm wavelength, in most instances of prostate cancer, the Raman light-scattering spectrum contained a series of distinct peaks in the range of 800–1500 cm^−1^, as shown in Figure 2. The Raman spectra were preprocessed by using spectrum smoothing, background subtraction and normalization (Figure 2a), and the same combination of methods without normalization (Figure 2b). These Raman shifts have been identified as follows: 898 cm^−1^ β for glucose [16]; 1005, 1032, and 1209 cm^−1^ for phenylalanine [17,18,19]; 1298 cm^−1^ for fatty acids [20]; 1400 cm^−1^ for glutamic acid [16]; and 1441 cm^−1^ for cholesterol and its esters [16,21]. Depending on the type of data processing, we can observe a different number of Raman peaks, but the PLS model can choose less obvious spectral features to discriminate the classes.

Although spectra of the hyperplasia regions and PC specimens had mostly matching Raman shifts, they showed a significant difference in luminescence intensity (in the spectral window of 350–500 cm^−1^). Another key distinguishing feature between the PC (prostate cancer) and BPH (benign prostatic hyperplasia) groups observed in Raman spectra is the differences in peak intensities of cholesterol, phenylalanine, glutamic acid, and β-glucose (Figure 3). The acquired spectral data were preconditioned, to reduce noise by Savitzky–Golay filtering and remove background radiation components by baseline correction with asymmetric least squares (baseline ALS). The data were normalized to the standard deviation (SNV—Standard Normal Variate). The obtained relative intensities were averaged over the PC and BPH groups.

Using the Raman spectra of prostate cancer and hyperplasia from a small number of samples as an example, two models of data separation by the PLS-DA method with dif-ferent data preprocessing were constructed. The analysis of the models and their accuracy of class discrimination (accuracy, sensitivity and specificity of adenocarcinoma di-agnosis) was performed. Model analysis includes: the number of latent variables, their intensity and noise level, variable importance in projection (VIP). VIP scores summarize the influence of individual X variables (Raman shifts) on the PLS model. VIP scores give a measure useful to select what are the variables which contribute most to the y variance explanation (adenocarcinoma-1/hyperplasia-0). VIP scores-vector summarizes all latent variables for a given model and spectral data set [22].

The first PLS-DA model included following preprocessing: spectrum smoothing (Savitzky–Golay filtering with zero derivative and zero polynomial, only averaging the signal in the window over fifteen points), background subtraction (baseline ALS), and normalization (SNV). For the constructed PLS-DA model, an analysis of its behavior was carried out on the calibration and validation datasets (Figure 4). Figure 4a shows that it is necessary to select two latent variables, since a break in the curve occurs, the discrimination accuracy on the validation dataset drops (the number of misclassified samples—NMCCV increases). Latent variables (loading vectors) show the greatest scatter of spectral data between classes when orthogonal decomposition. The maximum modulus values are in good agreement with the spectral features of the preprocessed spectra. In this case, it is worth noting high-frequency noises of low intensity, despite the smoothing of the spectra. However, excessive smoothing of the spectra can lead to the loss of spectral information.

The first analytical model developed by the PLS-DA method enabled differentiation between the Raman spectra of prostate cancer and areas of hyperplasia with 100% accuracy. At the same time, for the validation dataset, the accuracy was 70–80%, depending on the degree of specificity (50–67%), while the sensitivity was 70–86%. It is worth noting the form and intensity of the VIP spectrum in Figure 4e. It shows the three most significant spectral regions for class separation, but the spectrum itself is very noisy. This indicates that the noise contributes almost as much as the characteristic Raman bands. To eliminate this effect, a second model was proposed.

The preprocessing of the second PLS-DA model included only spectrum smoothing (Savitzky–Golay filtering) and background subtraction (baseline ALS) and without normalization, which results in an increase in the contribution of the noise component. The same analysis of the constructed PLS-DA model was performed (Figure 4c). The model shows its robustness even for five latent variables; the number of misclassified samples is reduced for the calibration and validation datasets. The stopping criterion for LV number determination is the form and the intensity of the latent variables. Figure 4d shows that the fifth variable mainly contains noise. Therefore, the second PLS-DA model contained only four latent variables.

The second PLS-DA model discriminated the Raman spectra of prostate cancer and hyperplasia with 100% accuracy, too. At the same time, for the validation dataset, the accuracy was 76–78%, depending on the degree of specificity (54–100%), while the sensitivity was 74–78%. Such strong subsidence in accuracy is associated with a small number of samples and an unequal number of samples in classes; there were fewer hyperplasias, which is reflected in large fluctuations in the specificity values for both models. Singular examples of coincidence between the spectral data for adenocarcinoma and hyperplasia nodes are attributed to the laser beam interacting with stromal regions or vascular walls, which have structural components common to all tissue types. It is important to note the VIP spectrum for the second model (Figure 4f). For the second model, the number of Raman peaks has increased, and, most importantly, there is practically no noise in the VIP spectrum. This fact shows the potential of this model to effectively discriminate on a larger number of samples, and is highly likely to reduce the discrimination difference between the calibration and validation datasets.

In isolated instances, in specimens of both groups, there were discovered glandular formations without well-structured cellular layers of epithelium, with signs of nuclear atypia and, yet, intact basal membrane. Immunohistochemical study of such cases revealed moderate expression of AMACR, as well as HMWCK (Figure 1), which was regarded as high-grade PIN (7.69%). However, due to the relatively small number of these observations, the result of their spectral analysis was of little merit to the given investigation for the lack of sufficiently convincing and credible data (*p* > 0.05).

### 2.3. Luminescence Spectroscopy with Laser Excitation at 532 nm Wavelength

At 532 nm laser excitation, optical response spectra of all tissue samples exhibited highly expressed luminescence, which significantly obscured visualization of the Raman light scattering, as illustrated by the curves of initial data for PC and BPH in Figure 5. Therefore, the data were mean centered with prior smoothing of the noise by 15-point averaging. The accuracy of PC discrimination using the PLS-DA model with one latent variable was 74% on the calibration dataset, and on the validation dataset, it drops to critically low values of 47–58%, depending on the ratio of sensitivity and specificity.

The laser excitation wavelength of 532 nm penetrates shallowly into the tissue, being absorbed in its near-surface layers [23], which excites a strong and spectral-wide luminescent response and, thus, does not allow to separate the classes of hyperplasia and adenocarcinoma.

### 2.4. Discussion

The outcomes of the given spectroscopic and morphological study are in full agreement with the fundamental theory of disrupted tissue and cell metabolism, which occurs in prostate cancer, contrary to the ordinary expression of the Warburg effect [24] observed in most cases of a malignant tumor. At the early stages of PC development, there is no aerobic glycolysis taking place in the cells. Instead, the main energy source becomes the lipids derived from androgens [25]. Furthermore, it has been demonstrated by Deep G. et al. that prostate cancer cells overexpress certain markers that play a central role in initiating lipid synthesis de novo [26]. These include fatty acid synthase (FASN), protein 1—binding the regulatory elements of sterol (SREBP1) and stearoyl-CoA desaturase, and others. Stearoyl-CoA desaturase, in particular, is an enzyme essential in forming monounsaturated fatty acids out of larger saturated ones.

The results of spectroscopic investigation provide clear evidence of many-fold higher concentrations of cholesterol and its esters (1441 cm^−1^), as well as fatty acids (1298 cm^−1^) in tumor cells of adenocarcinoma, compared to those of nodular hyperplasia. The level of β-glucose in adenocarcinoma samples was five times higher than in BPH (Figure 3), which might be related to lower rate utilization of glucose by tumor cells.

Spectral data also revealed an elevated level of glutamic acid (1400 cm^−1^) in instances of adenocarcinoma in contrast to BPH, which confirmed the assumption of the effect or role of glutamic acid in lipogenesis de novo and, consequently, in the stimulation of tumor growth [27].

Currently, numerous molecular studies have been conducted to confirm the hypothesis of identifying fatty acid, beta-glucose, glutamic acid, and cholesterol as biomarkers of prostate cancer [28,29,30].

The presence of characteristic Raman peaks of the identified individual molecules does not allow the diagnosis of cancer, since, for example, both cholesterol and beta-glucose can occur in benign hyperplasia. However, the presence of an ensemble of Raman peaks with increased intensity, inherent in fatty acid, beta-glucose, glutamic acid, and cholesterol, is a fundamental factor for the identification of prostate cancer.

## 3. Materials and Methods

### 3.1. Tissue Samples

We studied the prostate glands or their fragments removed during the operation of radical prostatectomy in patients (n = 105 patients, 40 benign tissues and 65 PC tissues) with suspected prostate cancer who were being treated at the Science and Research Institute of Uronephrology and Reproductive Health in I.M. Sechenov First Moscow State Medical University. Tissue fragments obtained during the standard procedure for cutting out the prostate for subsequent histological examination were preliminarily subjected to spectroscopic examination. The sizes of the tissue pieces for the spectral study were 1.0 cm × 1.0 cm × 0.5 cm. The fate of the peripheral zone of the prostate, suspected of cancer, as well as nodes of hyperplasia, visually having clear contours, were studied in detail. The local ethics committee of Sechenov University approved the study “preoperative and intraoperative diagnostics of prostate cancer using Raman-luminescence spectroscopy” (Protocol No. 19-20, 2020). The 117 spectra (532 nm laser) and 156 spectra (785 nm laser) were collected from tissue samples. After spectra registration from suspicious areas, the same samples were fixed in 10% buffered formalin solution, embedded in paraffin, and then subjected to microtomy, for further standard histological examination. Among the clinical data, the age of the patients, the level of total PSA in the blood serum, the volume of the prostate gland, and the presence of regional and distant metastases were assessed. Morphological examination included standard histological examination with hematoxylin and eosin, as well as immunohistochemical examination, using antibodies to AMACR and HMWCK. All observations of prostate cancer were graded according to the Gleason scale, as modified by Reference [31]. 

The overall distribution of the patients following the international staging classification of malignant tumors (pTNM—pathological Tumor-Node-Metastasis) is shown in Table 1.

### 3.2. Experimental Setup

The prostatic tissue samples were examined by histological methods—hematoxylin and eosin staining, and Masson’s trichrome staining, as well as immunohistochemical methods—HMWCK (high-molecular-weight cytokeratin) and AMACR (Alpha-Methylacyl-CoA Racemase). The registration and analysis of Raman scattering were carried out by utilizing two measurement systems. The first one consisted of three modules—a narrowband excitation laser source, LML-785.0RB-04 by PD-LD Inc. (Pennington, NJ, USA), producing radiation at 785 nm wavelength, with the probe output power of 200 mW; an optical filter unit, Raman probe RPB785 by InPhotonics (Norwood, MA, USA) with a diameter of the output beam in the focus of about 0.25 mm; and an imaging spectrometer Shamrock SR-500i by Oxford Instruments (Abingdon, UK) (spectral resolution of 1 cm^−1^) with integrated ANDOR DV-420A-OE camera, with the sensor matrix cooled to −70 °C, for detection of low-intensity optical signals. This scheme was employed to obtain spectral data in the 780–950 nm range with 60 s exposure time. The other system, Raman-luminescence spectrometer EnSpectr R532 by Enhanced Spectrometry, Inc. (San Jose, CA, USA), included a 1×-grating spectrograph, with no moving parts, rigidly joined with a 532 nm laser assembly. This instrument provided a spectral range of 540–680 nm at a spectral resolution of 4–6 cm^−1^. For a laser source with a radiation wavelength of 532 nm, the lens magnification was 10-fold. 

### 3.3. Mathematical Analysis of Prostate Spectral Data

The underlining of the spectral features of cancerous tissues was conducted by the method of projection onto latent structures (partial least squares) with linear discriminant analysis (PLS-DA). PLS is based on principal component analysis (PCA). PCA is a statistical technique used to simplify complex datasets and identify key variables in a multivariate dataset that can best explain the differences in observations. To reduce the size of spectral data, PCA is commonly used to extract a set of orthogonal principal components (PCs) that account for the maximum variance in the dataset for further diagnostics and characterization.PLS-DA can be successfully applied to multiclass classification problems by encoding the class membership of zeros and ones representing the group affinity in the appropriate Y-indicator matrix. The collected intensity characteristics of Raman spectra were used to construct a matrix, where each tissue specimen was assigned 0 or 1, depending on the classification of being intact or tumorous, respectively. PLS-DA uses the fundamental principle of PCA, but it additionally «rotates» the latent variables (LV) components for maximum covariance between spectral variation and group affinity, so that LVs explain diagnostically significant variations rather than the most noticeable variations in the spectral dataset. In most cases, this ensures the preservation of diagnostically significant spectral variations in the first few LVs.It should be noted that PCA-DA and PLS-DA have been repeatedly used to diagnose tissues based on the features of Raman spectra. At the same time, the discrimination accuracy using the PLC-DA is usually 3–6% higher than that of the PCA-DA [32,33,34].

Moreover, PLS-DA enabled the identification of spectral features of the classes associated with the presence of compounds and molecules, regarded as tumor markers localized in prostatic tissue. Prior to PLS-DA, the acquired data were preconditioned to reduce noise by Savitzky–Golay filtering [35] and remove background radiation component by baseline correction with asymmetric least squares (baseline ALS) [36]. For one of the PLS-DA model, the data were normalized to the standard deviation (SNV—Standard Normal Variate) [37]. SNV method performs a normalization of the spectra that consists of subtracting each spectrum by its own mean and dividing it by its own standard deviation. Then, PLS-DA models were built for the preprocessed data, to provide the means of PC and BPH class differentiation. During PLS-DA modeling, the behavior of constructed models was tested on “independent” data (validation dataset), using 10-fold cross-validation. All preprocessing and multivariate analysis of collected data were carried out online in TP^T^cloud. The matching of Raman peaks to particular classes of substances was accomplished with the help of a reference database of Raman spectra for biological tissues [16]. The statistical calculations were performed on a personal computer, using Microsoft Excel spread sheets and an application pack Statistica for Windows 7.0 by Dell Technologies Inc. (Round Rock, TX, USA).

All experimental output, including quantitative, anamnestic, clinical, laboratory, and instrumentation data, was handled by the method of variance statistics.

## 4. Conclusions

The difficulties of differential diagnosis of prostate cancer and precancerous processes contribute to the development and introduction into clinical practice of the latest diagnostic methods based on the use of high-tech and sensitive equipment. At present, reports on the use of laser optical spectroscopy in the diagnosis of a wide number of diseases are increasingly appearing in the world literature [5,38,39,40,41,42,43]. Compared to traditional imaging methods (Magnetic Resonance Imaging and Transrectal Ultrasound Diagnostics), optical methods have a number of unique advantages. First, optical radiation is non-ionizing, so it does not pose a health hazard, even for a long time of exposure. Second, optical measurements are based on the biochemical and morphological changes in the tissue under study. Third, the development of fiber-optic probes made it possible to include Raman spectroscopy in endoscopic equipment, which allows us to predetermine the histological process, as well as to determine the surgical edges of the resection and the presence of metastases as accurately as possible.

In this work, the integration of morphological and spectroscopic studies made it possible to determine the specifics of structural modifications in prostatic tissue during hyperplastic and tumorous transformations and to identify optical tumor markers—a form of distinct tissue “fingerprint”. The built analytical models of PLS-DA allowed us to divide the Raman spectra of prostate cancer and the spectra of hyperplasia sites with the accuracy up to 80% for validation dataset, while, for calibration datasets, the accuracy values were 100% for laser excitation and 785 nm.

The given experimental data make it evident that reproducible, highly accurate, and sensitive diagnosis of prostate cancer can be achieved by multimodal analysis of spectral data of biological fluids and tissues of oncological patients acquired with several spectrometer systems at different excitation wavelengths.

A promising direction is the use of volume Raman spectroscopy for multi-component spectral analysis of biological tissues and biological fluids of patients with cancer. In the future, we plan similar studies that will allow us to establish correlations between the development of a malignant neoplasm, changes in tissue metabolism, and changes in the biochemical composition of the patient’s biological fluids. 

In addition, a field that is actively developing and can complement Raman spectroscopy is surface-enhanced Raman spectroscopy (SERS). The search and detection of circulating cancer cells or molecular biomarkers in the blood of cancer patients, using SERS sensors, are actively developed by the authors of this work, and, in the future, these sensors will be adapted for the diagnosis of prostate cancer.

All of the above shows the potential prospects of establishing a novel methodology of PC research involving technologies from various fields of science, in order to improve the early diagnosis of prostate cancer and to gain further insight into the nature of its cancerogenesis.

## Figures and Tables

**Figure 1 molecules-26-01961-f001:**
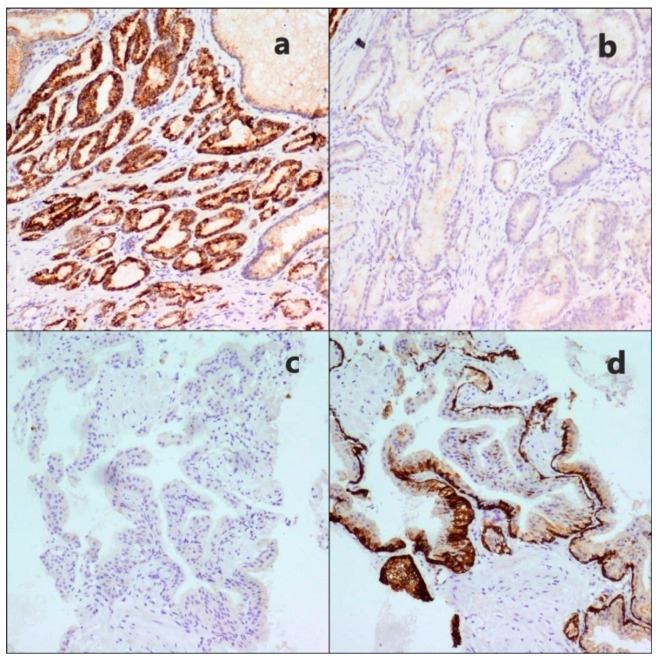
Comparison of immunohistochemical characteristics of prostate cancer and benign hyperplasia. (**a**) Diffuse intense cytoplasmic staining of the neoplastic glands of adenocarcinoma with Alpha-Methylacyl-CoA Racemase (AMACR), ×40. (**b**) Negative basal cell staining with high-molecular-weight cytokeratin (HMWCK) of adenocarcinoma, ×40. (**c**) Benign hyperplasia tissue with negative expression of AMACR, ×20. (**d**) HMWCK expression in benign prostatic hyperplasia, ×20.

**Figure 2 molecules-26-01961-f002:**
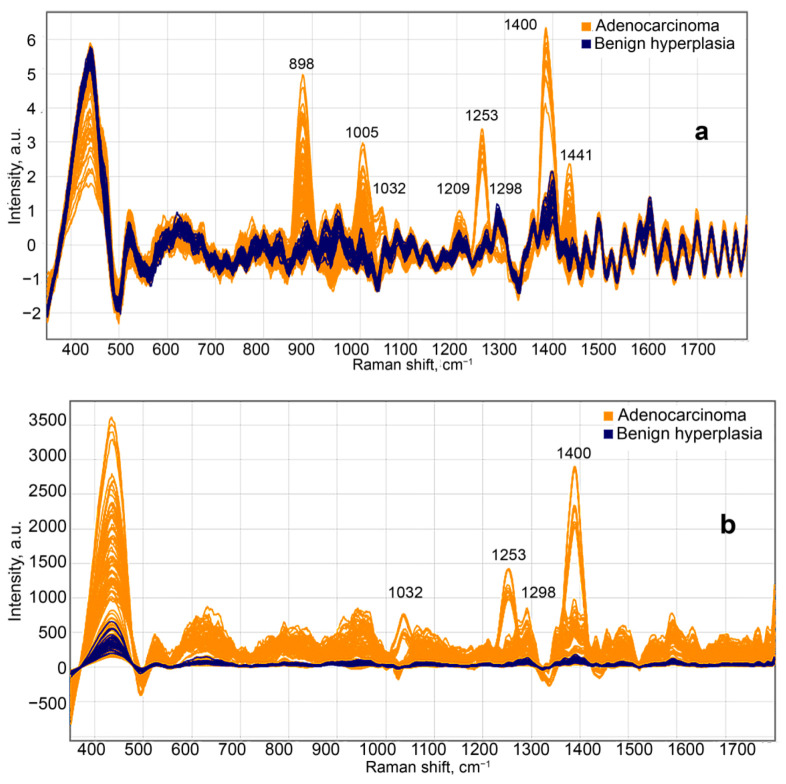
Preprocessed Raman spectra, using spectrum smoothing, background subtraction, and normalization (**a**) and the same combination of methods without normalization (**b**) from the regions of acinar adenocarcinoma and benign hyperplasia in prostatic tissue, produced by laser excitation of 785 nm wavelength.

**Figure 3 molecules-26-01961-f003:**
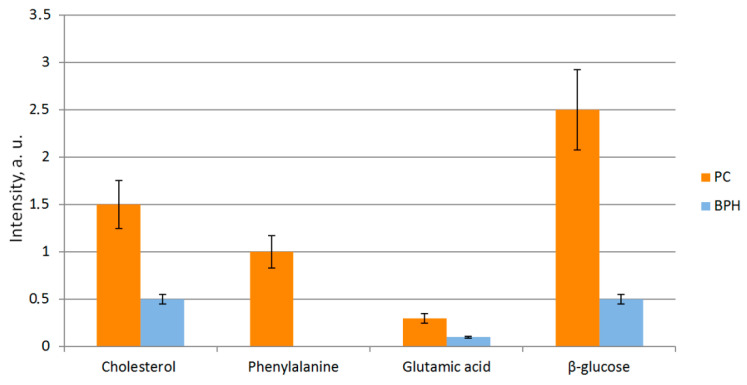
Relative intensities of Raman shifts, with error bars denoting several substances identified in PC and BPH samples. Level «0» corresponds to the level of noise in the spectrometer, which does not allow detecting signals that are lower in intensity than these noises.

**Figure 4 molecules-26-01961-f004:**
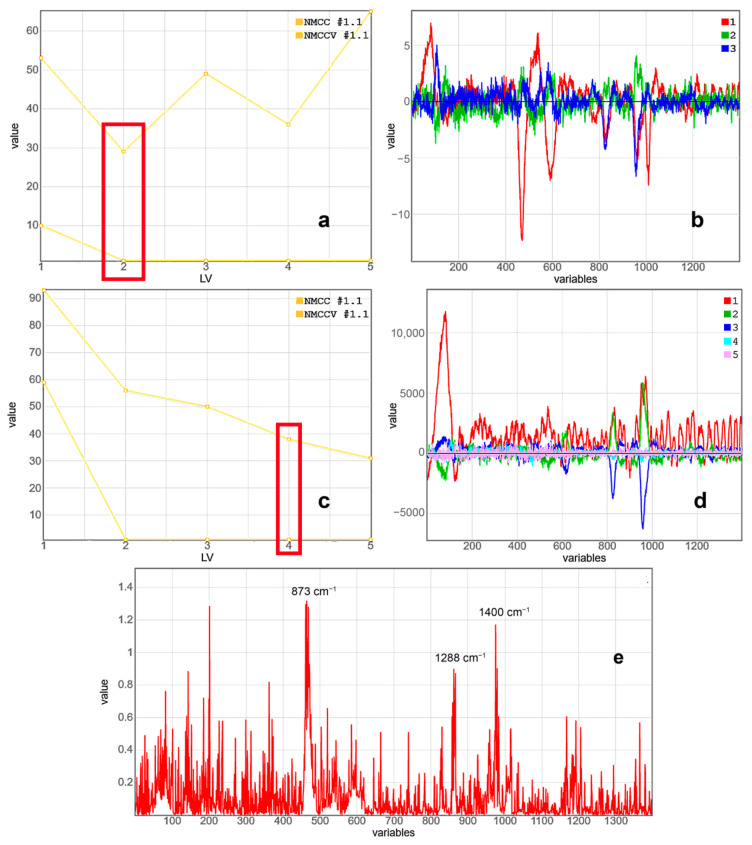
(**a**) First projections onto latent structures data analysis (PLS-DA) model. Dependence of the number of misclassified (NMC) samples from the number of latent variables (LVs) for calibration (NMCC) and validation (NMCCV) data-sets. (**b**) First PLS-DA model. Loading vectors 1 (red), 2 (green) and 3 (blue) of the PLS-DA model used for prostate cancer (PC) and benign prostatic hyperplasia (BPH) class differentiation. (**c**) Second PLS-DA model. Dependence of the number of misclassified (NMC) samples from the number of latent variables (LV) for calibration (NMCC) and validation (NMCCV) data-sets; (**d**) Second PLS-DA model. Loading vectors 1 (red), 2 (green), 3 (blue), 4 (turquoise), and 5 (pink) of the PLS-DA model used for PC and BPH class differentiation. (**e**) VIP spectrum of first PLS-DA model (Savitzky–Golay filtering, baseline asymmetric least squares (baseline ALS), and Standard Normal Variate (SNV)) for PC and BPH samples discrimination. (**f**) VIP spectrum of second PLS-DA model (Savitzky–Golay filtering, Baseline ALS, and SNV) for PC and BPH samples discrimination.

**Figure 5 molecules-26-01961-f005:**
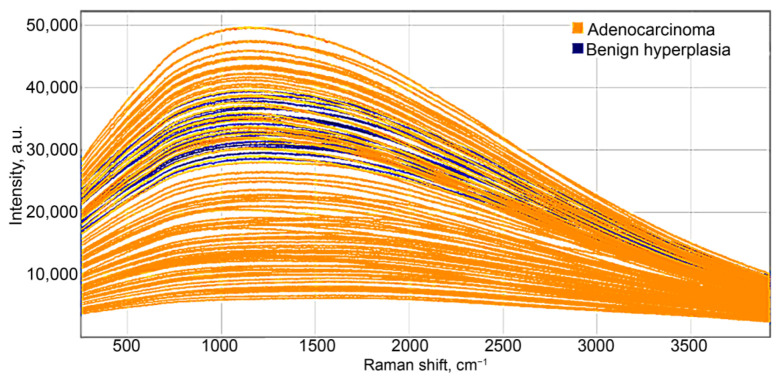
Smoothed luminescence spectra of PC and BPH excited by laser excitation at 532 nm wavelength.

**Table 1 molecules-26-01961-t001:** Clinical and morphological characteristics of patients with PC and BPH.

Age	Prostate Cancer (n = 65)	BPH (n = 40)
<65	17 (26.2%)	24 (60%)
66–74	35 (53.8%)	13 (32.5%)
>75	13 (20%)	3 (7.5%)
**PSA level, ng/mL**		
<4	5 (7.7%)	37 (92.5%)
4–10 h	42 (64.6%)	3 (7.5%)
>10	18 (27.7%)	0
**Volume, cm^3^**		
<40	12 (18.5%)	10 (25%)
40–60	38 (58.4%)	27 (67.5%)
>60	15 (23.1%)	3 (7.5%)
**pTNM**		
pT1	Not found	
pT2	43 (66.1%)	
pT3	22 (33.8%)	
pT4	Not found	
**Gleason score**		
<6	24 (36.9%)	
7 (3 + 4)	15 (23.07%)	
7 (4 + 3)	14 (21.5%)	
8	5 (7.7%)	
9–10	7 (10.7%)	

## Data Availability

The data presented in this study are available on request from the corresponding author. The data are not publicly available due to privacy.

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
