# Peer review of "Using the Method of “Optical Biopsy” of Prostatic Tissue to Diagnose Prostate Cancer"

_molecules, 2021, doi:10.3390/molecules26071961_

Round 1

Reviewer 1 Report

Please add the error bars with s.e.m. to figure 3

Check English for grammar

Add more relevant references

Expand the conclusions a bit more. broader implications of this study

Author Response

We are grateful for the valuable comments of the Reviewer.

Comment: Please add the error bars with s.e.m. to figure 3.

Answer: Done. We added the error bars in Figure 3.

Comment: Check English for grammar.

Answer: Done.

Comment: Add more relevant references.

Answer: Relevant references on the research of prostate cancer tissues using Raman spectroscopy have been added to the text of the article [doi:10.1364/boe.9.004294, doi:10.1117/1.jbo.23.12.121613, doi:10.1111/bju.14199, doi:10.1002/jbio.201700166]. The added papers were released in 2017-2018. In all works, prostate tissues are examined. The possibilities of Raman spectroscopy with excitation at different wavelengths of laser radiation: 633, 785, 1064 nm were investigated. In the presented works, the differences in the average spectra of cancer and control groups were analyzed, or the method of principal components was applied. In our work, for data analysis, we used the method of projections onto latent structures - PLS (PCA was used at the preliminary stage), which does not show the greatest differences of data between classes, but the most informative bands for separating classes (i.e., it is not necessary that there are maximum differences in the intensities of the spectra). Based on this analysis, the most informative Raman bands for separation (VIP variables) were identified, which were subsequently interpreted as possible biochemical (molecular) and structural changes in tissues.

Comment: Expand the conclusions a bit more. broader implications of this study

Answer: In the «Conclusion» section, information was added about the benefits of using Raman spectroscopy for the diagnosis of prostate cancer: «The difficulties of differential diagnosis of prostate cancer and precancerous processes contribute to the development and introduction into clinical practice of the latest diagnostic methods based on the use of high-tech and sensitive equipment. At present, reports on the use of laser optical spectroscopy in the diagnosis of a wide number of diseases are increasingly appearing in the world literature [doi:10.3390/ijms160714554, doi:10.1177/153303460300200407, doi: 10.1039/c0an00527d, doi: 10.1002/lsm.20653, doi: 10.1046/j.0022-202X.2004.22208.x, doi:10.1038/s41598-019-56308-y, doi:10.3390/ijms21144828]. Compared to traditional imaging methods (MRI, Transrectal Ultrasound Diagnostics), optical methods have several unique advantages. First, optical radiation is non-ionizing, so it does not pose a health hazard, even for a long time of exposure. Second, optical measurements are based on the biochemical and morphological changes in the tissue under study. Third, the development of fiber-optic probes made it possible to include Raman spectroscopy in endoscopic equipment, which allows us to pre-determine the histological process, as well as to determine the surgical edges of the resection and the presence of metastases as accurately as possible. In addition, the direction of using volumetric Raman spectroscopy to conduct a multicomponent spectral analysis of biological tissues and biological fluids of patients with oncological diseases is promising. In the future, we plan similar studies that will allow us to establish correlations between the development of a malignant neoplasm, changes in tissue metabolism, and changes in the biochemical composition of the patient's biological fluids. A field that is actively developing and can complement Raman spectroscopy is surface-enhanced Raman spectroscopy (SERS). The search and detection of circulating cancer cells or molecular biomarkers in the blood of cancer patients using SERS sensors are actively developed by the authors of this work and in the future, these sensors will be adapted for the diagnosis of prostate cancer».

Reviewer 2 Report

Artemyev et al. present a PLS-DA method for the purpose of diagnosing prostate cancer. In this manuscript, the authors proposed an anlytical discrimination models of Raman spectra using both 532 nm and 785 nm as excitation wavelength. By matching the Raman fingerprints, cellular metabolites exclusively occurring in the tissue of malignant prostate tumor samples were compared. This is a significant field to explore and more accurate methods are expected to improve the patients' clinical outcome. However, this manuscript in the current version needs improvements in their analytical statistics prior to its publications.

Major issue: the authors quantified the Raman intensity at indicated wavelength to indicate their potential as a biomarker for the diagnosis of prostate cancer. However, it lacks of statistic significance and sample analysis. In addition, what is the selectivity and specificity of this biomarkers? The authors need to address this issue since this work focuses on disease diagnosis.

Minor issues: The authors need to mind their utilization of punctuation mark throughout the manuscript. 

Author Response

We are grateful for the valuable comments of the Reviewer.

Comment: The authors quantified the Raman intensity at indicated wavelength to indicate their potential as a biomarker for the diagnosis of prostate cancer. However, it lacks of statistic significance and sample analysis.

Answer: The authors agree with the expert's arguments, but this paper demonstrates the potential use of Raman spectroscopy for monitoring quantitative changes in tissue metabolites. Undoubtedly, a further set of materials and further research will contribute to the formation of a more accurate method for assessing changes in the composition and content of tissue metabolites in the development of malignant prostate diseases.

Comment: In addition, what is the selectivity and specificity of this biomarkers? The authors need to address this issue since this work focuses on disease diagnosis.

Answer: In the «Discussion» section, information has been added on the identification of the molecules shown in Figure 3 as biomarkers of prostate cancer: «Currently, numerous molecular studies have been conducted to confirm the hypothesis of identifying fatty acid, beta-glucose, glutamic acid, and cholesterol as biomarkers of prostate cancer [doi:10.3390/molecules25071652, doi: 10.1002/jcb.10724, doi: 10.1101/220400].

The presence of characteristic Raman peaks of the identified individual molecules does not allow the diagnosis of cancer, since, for example, both cholesterol and beta-glucose can occur in benign hyperplasia. However, the presence of an ensemble of Raman peaks with increased intensity, inherent in fatty acid, beta-glucose, glutamic acid, and cholesterol, is a fundamental factor for the identification of prostate cancer.

Comment: The authors need to mind their utilization of punctuation mark throughout the manuscript.

Answer: Done

Reviewer 3 Report

This study provides a possible ability of Raman spectral analysis for discrimination of adenocarcinoma from hyperplasic of the prostate.

(Minor comment) Labels and b in Figures 4 and 5 should be placed appropriately. 

Author Response

Comment: Labels and b in Figures 4 and 5 should be placed appropriately.

Answer: Thank you so much for your comment. We have placed the labels in Figures 4 and 5 correctly.

Reviewer 4 Report

The authors present Raman spectroscopy method to detect Prostrate Cancer (PC) in n = 105 patients (40 benign tissues and 65 PC tissues). Statistical analysis was performed using PLS-DA to classify the samples into cancer and benign groups. The manuscript might be improved by incorporating the following comments:

  1. Since only 117 spectra (532 nm laser) and 156 spectra (785 nm laser) were taken, it seems only 1 -2 spectra/tissue sample was obtained. How the region was selected from where the spectra were obtained?
  2. The size of the tissue, magnification of objectives used should be provided.
  3. All of these tissue will have high autofluorescence. How was it avoided?
  4. Some details about the PLS-DA should be mentioned.
  5. In Figure 3, how the relative intensities were calculated? Although both PC and BPH have Phenylalanine, in Figure 3, why is it shown for only PC?
  6. Why these molecules (beta-glucose, glutamic acid, cholesterol) were chosen? It is understood that all of these molecules will have overlapping peaks - how these molecules were identified unequivocally?
  7. All the figures need modification. It should be of high quality and should not be truncated.
  8. Figure 4 to 8 could be condensed to a single figure. What is the significance of Figure 8?

Author Response

We are grateful for the valuable comments of the Reviewer.

Comment: Since only 117 spectra (532 nm laser) and 156 spectra (785 nm laser) were taken, it seems only 1 -2 spectra/tissue sample was obtained. How the region was selected from where the spectra were obtained?

Answer: Since the spot size of the laser radiation was large (for a laser spectrometer with a laser excitation wavelength of 785 nm, the spot size in the focus was 0.25 mm), there was no need to measure at different spatial points due to the averaging of the optical response signal during the measurement over the macroscopic area. The spectra were recorded in the areas marked by oncologists and verified by histological examination. The diagnosis was confirmed in advance with a standard histological examination on the material obtained by puncture biopsy.

Comment: The size of the tissue, magnification of objectives used should be provided.

Answer: The sizes of the tissue pieces for the spectral study were: 1.0x 1.0x 0.5 cm. For a laser source with a radiation wavelength of 532 nm, the lens magnification was 10-fold. The spectrometer with a wavelength of 785 nm used a fiber probe with a diameter of the output beam in the focus of about 0.25 mm.

Comment: All of these tissue will have high autofluorescence. How was it avoided?

Answer: Indeed, on a spectrometer with a laser wavelength of 532 nm, autofluorescence was extremely high in intensity, as a result, the Raman scattering signal was not detected. On a spectrometer with a laser wavelength of 785 nm, autofluorescence was small, since such low-frequency radiation does not excite it so effectively and tissue fluorophores do not absorb this wavelength. In addition, in the process of mathematical processing of the spectra, the baseline function was used, which allowed us to subtract a wide fluorescence profile and better visualize narrow Raman peaks.

Comment: Some details about the PLS-DA should be mentioned.

Answer: In the «Materials and methods» section, we have included a description of the features of the PLS-DA method: «The underlining of the spectral features of cancerous tissues was carried out by the method of projection onto latent structures (partial least squares) with linear discriminant analysis (PLS-DA). PLS is based on principal component analysis (PCA). PCA is a statistical technique used to simplify complex datasets and identify key variables in a multivariate dataset that can best explain differences in observations. To reduce the size of spectral data, PCA is commonly used to extract a set of orthogonal principal components (PCs) that account for the maximum variance in the dataset for further diagnostics and characterization.

 PLS-DA can be successfully applied to multiclass classification problems by encoding the class membership of zeros and ones representing the group affinity in the appropriate Y-indicator matrix. The collected intensity characteristics of Raman spectra were used to construct a matrix, where each tissue specimen was assigned 0 or 1, depending on the classification of being intact or tumorous, respectively. PLS-DA uses the fundamental principle of PCA, but additionally «rotates» the latent variables (LV) components to maximize covariance between spectral variation and group affinity, so that LVs explain diagnostically significant variations rather than the most noticeable variations in the spectral dataset. In most cases, this ensures the preservation of diagnostically significant spectral variations in the first few LVs.

 It should be noted that PCA-DA and PLS-DA have been repeatedly used to diagnose tissue diseases based on the features of Raman spectra. At the same time, the discrimination accuracy using the PLC-DA is usually 3-6% higher than that of the PCA-DA [doi:10.1039/c1an15296c, doi:10.1155/2016/1603609, doi:10.3892/mco.2014.473]»

Comment: In Figure 3, how the relative intensities were calculated?

Answer: The acquired spectral data were preconditioned to reduce noise by Savitzky-Golay filtering and remove background radiation components by baseline correction with asymmetric least squares (baseline ALS). The data were normalized to the standard deviation (SNV - Standard Normal Variate). The obtained relative intensities were averaged over the PC and BPH groups. This information about the intensity calculation is shown in Fig.3 added to the article.

Comment: Although both PC and BPH have Phenylalanine, in Figure 3, why is it shown for the only PC?

Answer: Undoubtedly, Phenylalanine is also present in BPH, but apparently its content is much lower and therefore we do not have enough sensitivity of the measuring CCD matrix to identify it among the instrument noise. Conventionally, level 0 in Figure 3 corresponds to the level of noise in the spectrometer, which does not allow detecting signals that are lower in intensity than these noises.

Comment: Why these molecules (beta-glucose, glutamic acid, cholesterol) were chosen? It is understood that all of these molecules will have overlapping peaks - how these molecules were identified unequivocally?

Answer: In the scientific literature, in studies devoted to the identification of biological molecules in cancer tissues using Raman spectroscopy, individual Raman peaks were identified, namely β-glucose, glutamic acid, fatty acids, cholesterol, phenylalanine: β-glucose (898 cm-1)[doi: 10.1038/sj.bjc.6603102],glutamic acid(1400 cm-1) [doi: 10.1038/sj.bjc.6603102], fatty acids (1298 cm-1) [doi: 10.1016/j.saa.2004.11.017],  cholesterol and its esters (1441 cm-1) [ doi: 10.1002/cyto.990140303], phenylalanine (1005,1032, 1209 cm-1) [doi: 10.1002/jemt.20229, doi: 10.1002/jrs.882, doi: 10.1098/rsif.2004.0008].

Comment: All the figures need modification. It should be of high quality and should not be truncated.

Answer: Done

Comment: Figure 4 to 8 could be condensed to a single figure.

Answer: Thank you for your comment. Figure 4,5,6,7 in the new version of the publication, we have combined how the images obtained in one spectroscopic study using a spectrometer with a laser wavelength of 785 nm. Figure 8 stands out from the total number, since the spectra shown on it were obtained on another spectrometer with a laser wavelength of 532 nm.

Comment: What is the significance of Figure 8?

Answer: Figure 8 is given in this article in order to demonstrate the presence of high-intensity autofluorescence when using laser radiation with a wavelength of 532 nm, which is not very useful for the diagnosis of prostate cancer and is inferior in information content to Raman scattering. However, for a number of tissues (breast, skin, brain), the wavelength of 532 nm can be informative due to the high content of resonant metabolites (carotenoids, phospholipids, etc.). Specifically for the diagnosis of prostate cancer, this wavelength turned out to be uninformative and this gives additional understanding to the world community that for further development of the method and its application in clinical practice, it is necessary to use low-frequency laser radiation at the edge of the visible range (785 nm) or in the IR region (1064 nm).

Round 2

Reviewer 2 Report

The authors partially addressed the concerns raised in the previous review cycle. The editor may make the decision per his or her judgement.

Reviewer 4 Report

The authors have incorporated all the comments.